

# Associations of *IGF2* and *DRD2* polymorphisms with laying traits in Muscovy duck

Qiao Ye, Jiguo Xu, Xinfeng Gao, Hongjia Ouyang, Wei Luo and Qinghua Nie

National-Local Joint Engineering Research Center for Livestock Breeding, College of Animal Science, South China Agricultural University, Guangzhou, Guangdong, China
Key Lab of Chicken Genetics, Breeding and Reproduction, Ministry of Agriculture and Guangdong Provincial Key Lab of Agro-animal Genomics and Molecular Breeding, South China Agricultural University, Guangzhou, Guangdong, China

## ABSTRACT

Insulin-like growth factor 2 (IGF2) and dopamine receptor 2 (DRD2) play important roles in ovarian follicular development. In this study, we analyzed tissue-specific expression of the Muscovy duck *IGF2* and *DRD2* genes and cloned those genes transcripts. Polymorphisms in these genes were tightly linked with egg production traits and both genes were highly expressed in the ovary. Moreover, we identified five single nucleotide polymorphisms (SNPs) for *IGF1* and 28 for *DRD2*. Mutations A-1864G and C-1704G of *IGF2* were positively correlated with increased egg laying at 59 weeks (E59W) ($P < 0.05$). The C+7T and C+364G mutations of *DRD2* were highly and significantly associated with first-egg age (FEA) and egg numbers at 300 days (E300D) ($P < 0.01$). Moreover, C+3301G and C+3545G of *DRD2* were highly significantly associated with FEA, E59W and E300D ($P < 0.01$). Other mutations were positively associated with FEA or E300D or E59W ($P < 0.05$). These data suggest specific roles for *IGF1* and *DRD2* polymorphisms in egg production in Muscovy ducks.

## INTRODUCTION

Muscovy ducks are an excellent breed species because of their rapid growth, crude feed tolerance and high-quality meat. Although these ducks are raised on a large scale in China, low production performance affects the economic interests of farmers. Breeders have been looking for ways to improve Muscovy ducks egg production. In recent years, with the rapid development of genome sequencing technologies, molecular marker breeding and transgenic breeding technology have gradually become the mainstream of breeding. Traditional breeding mainly depends on breeding experience, which has a lot of unpredictability. Furthermore, molecular breeding can significantly improve breeding efficiency and shorten the breeding period, so using molecular marker breeding has a huge advantage in breeding. Nowadays, molecular markers are widely used in poultry breeding, such as green shell egg related molecular markers (*Wang et al., 2013*) and egg production

Corresponding author
Qinghua Nie, nqinghua@scau.edu.cn

related molecular markers (*Han, An & Du, 2014*). Due to the great prospects of molecular markers in breeding, using molecular markers to selecting high laying performance Muscovy ducks is a good decision.

Our research focuses on egg production related molecular markers that can be used to improve egg production for the Muscovy duck. Few researchers have paid attention to egg production traits in Muscovy ducks, which makes our research more meaningful. The first egg age (FEA), egg numbers at 300 days (E300D), and egg numbers at 59 weeks (E59W) are important traits in Muscovy ducks breeding. Muscovy ducks egg peak time is from 35 weeks to 53 weeks, and 59 weeks is the last stage of laying. Three hundred days is the peak time of laying, and 59 weeks is the end time of laying in Muscovy ducks, which covers most of the egg laying period. Therefore, we focus on FEA, E300D and E59W instead of egg production at other time points as important traits.

Insulin-like growth factor 2 (IGF2) plays a key roles in animal growth differentiation and proliferation (*Kaneda et al., 2007*), as well as reproduction and the regulation of ovarian follicle development. In mammals, *IGF2* is highly expressed in the dominant follicle supporting key functions for follicular development (*Mao et al., 2004*). *IGF2* may affect prolificacy in sows and cattle (*Stinckens et al., 2010*; *Aad, Echternkamp & Spicer, 2013*), and *IGF2* may regulate ovarian development through follicle-stimulating hormone (FSH) (*Baumgarten et al., 2015*). But little research on the regulation of ovarian development by *IGF2* has been conducted in birds. The present study is the first to report that IGF2 may be associated with ovarian development. Dopamine (DA) is an essential neurotransmitter and exists in the nerve center and its peripheral tissue. Dopamine receptor 2 (DRD2) may assist with the secretion of reproductive hormones through follicle-stimulating hormone (FSH) and luteinizing hormone (LH) in chicken (*Youngren et al., 1996*; *Youngren, Chaiseha & El, 1998*). Association studies between single nucleotide polymorphisms (SNPs) of *IGF2* and *DRD2* and reproduction traits have been carried out in poultry (*Xu et al., 2011b*; *Xu et al., 2011a*; *Wang et al., 2014a*; *Wang et al., 2014b*; *Zhang et al., 2015*; *Zhu et al., 2015*). However, until now very few studies have focused on the relevance of these genes to egg production in Muscovy ducks. Therefore, we aim to identify SNPs in these genes, and to reveal their associations with reproduction traits in Muscovy ducks. We hope these molecular markers may help to improve the production performance of Muscovy ducks in breeding.

## MATERIALS AND METHODS

### Ethics statement

Ethical approval for all animal experiments was granted by the Animal Care Committee of South China Agricultural University (Guangzhou, People's Republic of China) with approval number 20131019002.

### Sample collection

Eight hundred white Muscovy ducks were offered by Wens Nanfang Poultry Breeding company (Yunfu, Guangdong, China) which were in the same run. All Muscovy ducks were reared under identical conditions of management and feeding. Ducks were maintained

outside on the ground from four to 12 weeks of age, after which they were transferred to individual cages in a semiconfined house. Feed was provided by Wens (Xinxing, Guangdong, China). The first egg age (FEA), egg numbers at 300 days (E300D), and egg numbers at 59 weeks (E59W) were recorded for each female duck. Genomic DNA from each individual at 59 weeks was isolated from 0.5 ml blood stored with EDTA as an anticoagulant, using E.Z.N.A NRBC Blood DNA Kit (Omega, Norcross, GA, USA) according to the manufacturer's instructions.

## RNA isolation and cDNA synthesis

Muscovy duck tissues including pituitary, brain, lung, abdominal fat, liver, ovary, subcutaneous fat, spleen, kidney, leg muscle, hypothalamus, cerebellum, heart and breast muscle used for expression pattern analysis of the *IGF2* and *DRD2* genes, were sampled at first egg age. These ducks were raised under the same conditions, but in different batches from the eight hundred Muscovy ducks mentioned above. Total RNA was isolated from tissues using a TRIZOL Reagent kit (TaKaRa, Dalian, China) according to the manufacturer's protocol. RNA quality was evaluated by 2% agarose gel electrophoresis and then was reverse transcribed using Takara reverse transcription Kit (TaKaRa, Dalian, China) according to the manufacturer's instructions. The cDNA was used as template to amplify the coding region of the *IGF2* and *DRD2* genes from Muscovy duck.

## Cloning of Muscovy duck *IGF2* and *DRD2* genes

The Muscovy duck *IGF2* and *DRD2* genes were identified using Mallard duck gene sequences as a reference (Gene Bank accession No. XM_005019778 and XM_013109685). Primers were designed to amplify the coding regions of Muscovy duck *IGF2* and *DRD2* using Primer 5.0 (Primer IGF2-CDS and Primer DRD2-CDS; Table S1). PCR amplifications were conducted in a final volume of 50 µl with 2 µl cDNA, 25 µl 2 × Easy Taq SuperMix (TransGen, Beijing, China), and 0.5 µl each pair of primers, and 22 µl double distilled H$_2$O. Optimum PCR amplification conditions were programmed as pre-denaturation at 94 °C for 3 min, followed by 35 cycles of denaturation at 94 °C for 30 s, annealing at 58 °C for 30 s, and extension at 72 °C for 30 s, and a final extension at 72 °C for 10 min. The PCR products were evaluated by electrophoresis using a 2% agarose gel and then gel purified using a HiPure Gel Pure DNA kit (TransGen, Beijing, China). The amplified fragments were cloned into pMD-18T vector (TaKaRa, Dalian, China), and sequenced by Majorbio, Shanghai, China. Sequence alignment and phylogenetic trees are constructed using MEGA5.

## Expression pattern analysis of *IGF2* and *DRD2* mRNA

Total mRNA from 14 different tissues was extracted to investigate the mRNA expression profiles of Muscovy duck *IGF2* and *DRD2* genes using real-time qPCR. Muscovy duck *β-actin* gene was used as the internal reference gene. Primers for the *IGF2,* DRD2 and *β-actin* genes were designed using Primer 5.0 (Primer *β-actin*-duck, IGF2-Q and DRD2-Q; Table S1). The qPCR was performed using a standard SYBR Premix Ex Taq II (TaKaRa, Dalian, China) on a BioRad CFX96 Real-Time PCR Detection System (Bio-Rad, Hercules, CA, USA) according to the manufacturer's protocol. The thermal cycling was 95 °C for 2

min, followed by 39 cycles of 95 °C for 15 s, 60 °C for 30 s, 72 °C for 30 s, and final cycle of 72 °C for 7 min. Relative expression of *IGF2* and *DRD2* genes was calculated relative to the expression of *β-actin*. Real-time PCR data were analyzed using the $2^{-\Delta\Delta Ct}$ method.

### SNPs detection by sequencing

We designed 7 primers to identify potential SNPs of *IGF2* and *DRD2* (Primer IGF2-P1, IGF2-P2, DRD2-P1, DRD2-P2, DRD2-P3, DRD2-P4 and DRD2-P5; Table S1). Twenty white Muscovy ducks were sampled and five individuals were selected as a mixed pool. PCR reactions were performed in a 50 µl final volume, containing 2 µl DNA, 25 µl 2 × Easy Taq SuperMix (TransGen Biotech, Beijing, China), 0.5 µl each pair of primers, and 22 µl double distilled $H_2O$. PCR parameters were 3 min at 94 °C followed by 37 cycles of 94 °C for 30 s, annealing temperature for 60 s, 72 °C for 30 s and a final extension at 72 °C for 10 min. PCR products were evaluated by electrophoresis using 2% agarose gel and sequenced as described above. SNPs were identified by the Seqman program of DNASTAR 7.1.0 software (DNASTAR, Inc., Madison, WI, USA).

### Genotyping and association analysis

The SNPs were genotyped in 800 female ducks with egg production records *via* sequencing. We designed 3 primers to genotyping SNPs of *IGF2* and *DRD2* (Primer IGF2-SNP, DRD2-SNP1 and DRD2-SNP2; Table S1). PCR reactions were identical to those used in SNP detection as described above. Genotypes were tested for Hardy-Weinberg equilibrium with the chi-square test. Linkage analysis was performed using Haploview software (*Barrett et al., 2005*). The associations between SNPs and egg production traits were calculated using the general linear model procedure of SAS v. 9.2 with the following model:

$$Y_{ij} = \mu + G_i + e_{ij}$$

where $Y_{ij}$ is the observed value of different egg production traits, $\mu$ is the overall population mean, $G_i$ is the effect of each genotype, and $e_{ij}$ is the random error. For each egg production trait, the least-squares mean was estimated and differences between the genotypes were analyzed using a Bonferroni adjustment for multiple comparisons. Difference with *P* value $\leq 0.05$ was considered to be significant in analyses.

## RESULTS

### Characterization of Muscovy duck *IGF2* and *DRD2* coding region

We obtained a 311-bp partial cDNA of the *IGF2* gene that was 98% and 95% identical to *Anas platyrhynchos* (XM_013191560.1) and *Anser cygnoides domesticus* (XM_005019778.2), respectively. We obtained the full-length cDNA of *DRD2* including a 52-bp 5′-untranslated region (UTR), an 1,104-bp open reading frame (ORF) containing 368 codons and a 294-bp 3′-UTR. The Muscovy duck *DRD2* cDNA sequence was 98% and 96% to *Anas platyrhynchos* (XM_013109686.1) and *Anser cygnoides domesticus* (XM_013187289.1), respectively. A phylogenetic tree constructed based on the *DRD2* gene also revealed that the Muscovy duck was closely related with both animals above (Fig. 1).
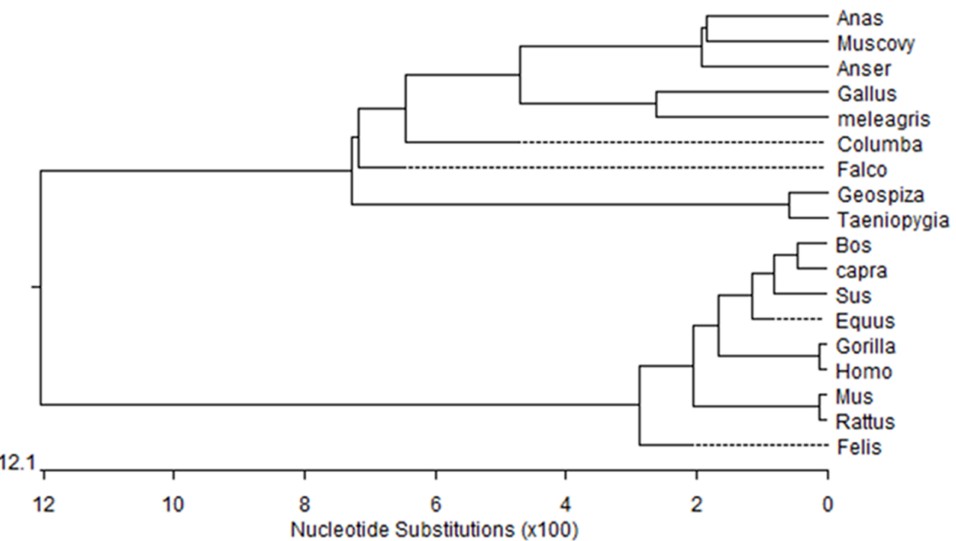

**Figure 1  Phylogenetic tree of Muscovy duck *DRD2* aligned amino acid sequences.** Orthologs were analyzed using ClustalW (*Goujon et al., 2010*).

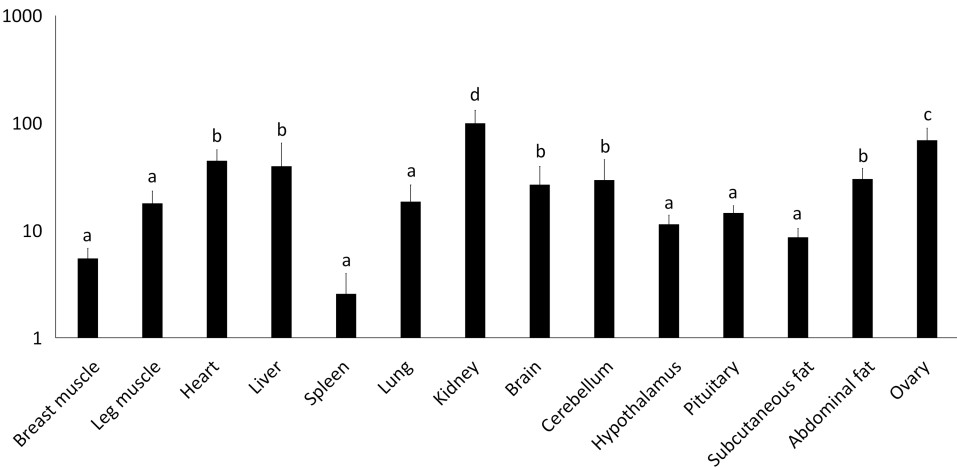

**Figure 2  Total mRNA expression of the *IGF2* gene in different tissues of the Muscovy duck.** The value in the $Y$ axis indicated $2^{-\Delta\Delta Ct}$ value.

## Tissue expression of *IGF2* and *DRD2* genes

We examined tissue-specific expression of IGF2 and found that it was expressed in most tissues. The highest expression levels were found in the kidney and ovaries (Fig. 2). *DRD2* expression was the highest in ovary, but it was also expressed in the cerebrum, cerebellum, hypothalamus and pituitary at lower levels. However, other tissues also had expression levels near detection limits including abdominal fat, sebum and breast and leg muscle. Expression in the spleen was negligible (Fig. 3).

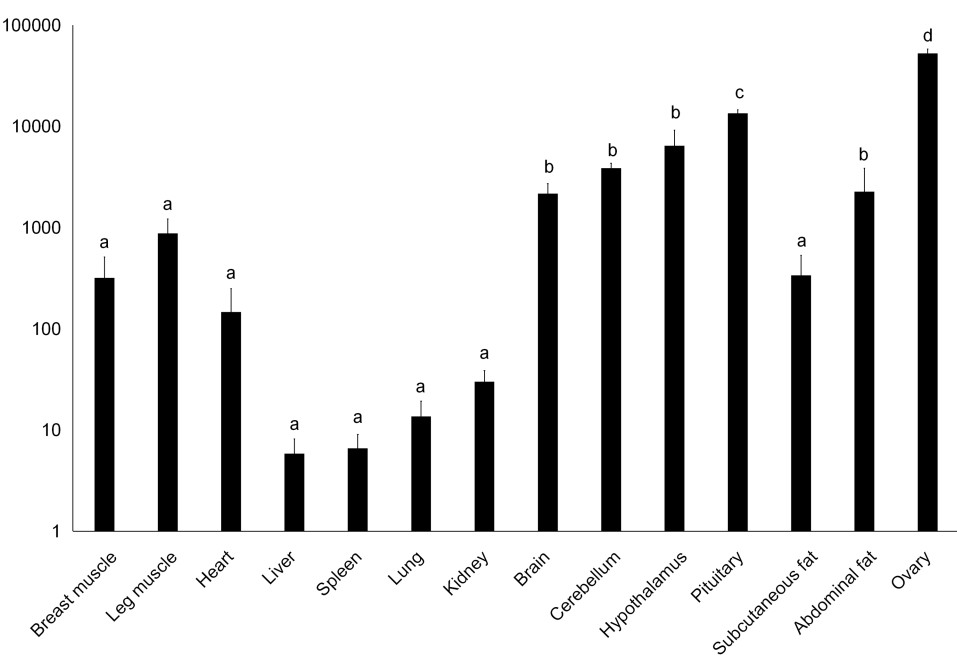

**Figure 3** **Total mRNA expression of the *DRD2* gene in different tissues of the Muscovy duck.** The value in the $Y$ axis indicated $2^{-\triangle\triangle Ct}$ value.

## Polymorphisms of *IGF2* and *DRD2* Genes

We identified 5 SNPs in the 5′ flanking region of *IGF2*, a level of one SNP per 449 bp on average. These SNPs were A-1864G, C-1704G, A-584G, A-227G and A-183G (Table 1). We found 28 SNPs in *DRD2* giving rise to one SNP per 317 bp on average. Among them, the SNP C+7T in exon 1 was a missense mutation resulting in a P to S amino acid change (Table 1). We selected 2 SNPs of *IGF2* and 11 SNPs of *DRD2*, based on mixed pool sequencing results which indicated that these were more likely to be associated with egg laying traits, for further association analysis.

## Association of *IGF2* and *DRD2* with egg production traits

Association analysis indicated that the A-1864G and C-1704G SNPs of *IGF2* gene were both significantly associated with E59W ($P < 0.05$) (Table 2), and linkage disequilibrium analysis indicated a high linkage block between A-1864G and C-1704G for *IGF2* (Fig. 4). Multiple comparisons of different genotypes showed that the AG genotype individuals of A-1864G had 6–7 eggs more than GG genotype individuals for E59W ($P < 0.01$). The GG genotype individuals of C-1704G had 7–8 eggs more than individuals with the CC genotype for E59W ($P < 0.05$).

Association analysis for *DRD2* gene further showed that C+7T and C+364G had highly significant associations with FEA and E300D ($P < 0.01$) and were significantly associated with E59W ($P < 0.05$) (Table 3). A+3489G, A+3484T and T+3428C were significantly associated with FEA and E300D ($P < 0.05$), and highly associated with E59W ($P < 0.01$). T+3423C and A+3262G were significantly associated with FEA ($P < 0.05$) and highly

**Table 1  SNPs identified in the *IGF2* and *DRD2* genes.**

| No. | Gene | SNPs[a] | Location[b] | Amino acid change |
|---|---|---|---|---|
| 1 | *IGF2* | A-1864G | 5′regulatory region | No |
| 2 | *IGF2* | C-1704G | 5′regulatory region | No |
| 3 | *IGF2* | A-584G | 5′regulatory region | No |
| 4 | *IGF2* | A-227G | 5′regulatory region | No |
| 5 | *IGF2* | A-183G | 5′regulatory region | No |
| 6 | *DRD2* | C-300G | 5′regulatory region | No |
| 7 | *DRD2* | A-251T | 5′regulatory region | No |
| 8 | *DRD2* | T-237G | 5′regulatory region | No |
| 9 | *DRD2* | A-194G | 5′regulatory region | No |
| 10 | *DRD2* | A-84G | 5′regulatory region | No |
| 11 | *DRD2* | C+7T | Exon 1 | Yes (P-S) (ccc-tcc) |
| 12 | *DRD2* | C+364G | Intron 1 | No |
| 13 | *DRD2* | A+476T | Intron 1 | No |
| 14 | *DRD2* | T+830G | Intron 1 | No |
| 15 | *DRD2* | T+3024C | Intron 1 | No |
| 16 | *DRD2* | A+3183C | Intron 2 | No |
| 17 | *DRD2* | A+3262G | Intron 2 | No |
| 18 | *DRD2* | C+3301G | Intron 2 | No |
| 19 | *DRD2* | T+3423C | Intron 2 | No |
| 20 | *DRD2* | T+3428C | Intron 2 | No |
| 21 | *DRD2* | A+3484T | Intron 2 | No |
| 22 | *DRD2* | A+3489G | Intron 2 | No |
| 23 | *DRD2* | C+3545G | Intron 2 | No |
| 24 | *DRD2* | T+6859G | Intron 5 | No |
| 25 | *DRD2* | T+6986C | Intron 5 | No |
| 26 | *DRD2* | T+7099C | Intron 5 | No |
| 27 | *DRD2* | T+7295C | Intron 5 | No |
| 28 | *DRD2* | T+7537C | Exon 6 | No |
| 29 | *DRD2* | C+7654G | 3′regulatory region | No |
| 30 | *DRD2* | T+8309G | 3′regulatory region | No |
| 31 | *DRD2* | A+8442G | 3′regulatory region | No |
| 32 | *DRD2* | T+8585C | 3′regulatory region | No |
| 33 | *DRD2* | A+8770G | 3′regulatory region | No |

**Notes.**

[a] SNPs means single nucleotide polymorphisms, referred to covered regions, the first nucleotide of the translation start codon was designated +1, with the next upstream nucleotide being −1.

[b] 5′ regulatory region = 5′ flanking and untranslated region; 3′ regulatory region = 3′ flanking and untranslated region.
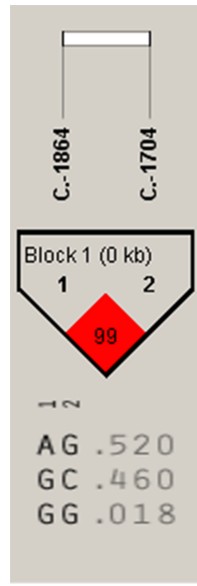

**Figure 4** **The linkage status of 2 identified SNPs in *IGF2* gene.** The color of block indicates the LD status of SNPs; deep red means high linkages between two SNPs.

**Table 2** **Association of 2 SNPs at *IGF2* gene with egg production traits in Muscovy duck.**

| SNPs[1] | Traits[2] | Least-squares mean ± SEM [3] | | | *P*-value |
|---|---|---|---|---|---|
| | | AA($n = 204$) | AG($n = 308$) | GG($n = 172$) | |
| A-1864G | FEA | $276.60 \pm 1.41^a$ | $275.67 \pm 1.15^a$ | $276.77 \pm 1.54^a$ | 0.8087 |
| | E59W | $75.35 \pm 1.92^a$ | $76.01 \pm 1.57^a$ | $69.18 \pm 2.10^b$ | 0.0251 |
| | E300D | $21.43 \pm 1.04^a$ | $21.91 \pm 0.85^a$ | $20.84 \pm 1.14^a$ | 0.7533 |
| | | CC($n = 158$) | CG($n = 310$) | GG($n = 216$) | |
| C-1704G | FEA | $276.50 \pm 1.61^a$ | $276.43 \pm 1.15^a$ | $275.72 \pm 1.37^a$ | 0.9067 |
| | E59W | $68.92 \pm 2.19^b$ | $75.33 \pm 1.56^a$ | $76.11 \pm 1.87^a$ | 0.0254 |
| | E300D | $20.97 \pm 1.19^a$ | $21.18 \pm 0.85^a$ | $22.33 \pm 1.01^a$ | 0.6050 |

**Notes.**

Data are summarized as means ± SEM.

[1] SNPs means single nucleotide polymorphisms, referred to covered regions, the first nucleotide of the translation start codon was designated +1, with the next upstream nucleotide being −1.

[2] FEA, first egg age; E59W, egg number at age 59 weeks; E300D, egg number at age 300 days.

[3] Values within a row with no common superscript differ significantly ($P < 0.05$) or are highly significant ($P < 0.01$).

significantly associated with E59W ($P < 0.01$). A+3183C was significantly associated with E59W ($P < 0.05$), and T+3024C has no significant association with any of the three egg production traits. Moreover, it was notable that C+3301G and C+3545G were highly significantly associated with FEA, E59W and E300D ($P < 0.01$). Multiple comparisons among different genotypes showed that the GG genotypes of C+3301G and C+3545G were advantageous for earlier egg laying and egg production. There were two high linkage blocks (C+7T and C+364G, A+3183C and A+3262G) for *DRD2* (Fig. 5).

**Table 3** Association of 11 SNPs at *DRD2* gene with egg production traits in Muscovy duck.

| SNPs[1] | Traits[2] | Least-squares mean ± SEM[3] | | | *P*-value |
|---|---|---|---|---|---|
| | | CC($n = 387$) | CT($n = 237$) | TT($n = 31$) | |
| C+7T | FEA | $272.95 \pm 0.90^c$ | $276.30 \pm 1.15^b$ | $295.94 \pm 3.17^a$ | <0.0001 |
| | E59W | $74.62 \pm 1.35^a$ | $75.34 \pm 1.73^a$ | $61.13 \pm 4.78^b$ | 0.0187 |
| | E300D | $22.90 \pm 0.71^a$ | $21.81 \pm 0.91^a$ | $8.42 \pm 2.51^b$ | <0.0001 |
| | | CC($n = 22$) | CG($n = 239$) | GG($n = 394$) | |
| C+364G | FEA | $297.00 \pm 3.80^a$ | $275.64 \pm 1.15^b$ | $273.79 \pm 0.90^b$ | <0.0001 |
| | E59W | $62.32 \pm 5.67^b$ | $77.15 \pm 1.72^a$ | $73.14 \pm 1.34^{ab}$ | 0.0193 |
| | E300D | $9.86 \pm 3.01^b$ | $22.37 \pm 0.91^a$ | $22.16 \pm 0.71^a$ | 0.0003 |
| | | TT($n = 130$) | TC($n = 160$) | CC($n = 410$) | |
| T+3024C | FEA | $276.27 \pm 1.75^a$ | $275.21 \pm 1.58^a$ | $277.19 \pm 0.98^a$ | 0.5547 |
| | E59W | $79.02 \pm 2.37^a$ | $72.81 \pm 2.13^{ab}$ | $72.77 \pm 1.33^b$ | 0.0594 |
| | E300D | $21.94 \pm 1.30^a$ | $21.37 \pm 1.17^a$ | $20.80 \pm 0.73^a$ | 0.7271 |
| | | CC($n = 143$) | AC($n = 182$) | AA($n = 375$) | |
| A+3183C | FEA | $276.84 \pm 1.67^a$ | $278.16 \pm 1.48^a$ | $275.67 \pm 1.03^a$ | 0.3816 |
| | E59W | $69.38 \pm 2.25^b$ | $72.85 \pm 2.00^{ab}$ | $76.21 \pm 1.39^a$ | 0.0301 |
| | E300D | $21.02 \pm 1.24^a$ | $19.98 \pm 1.09^a$ | $21.75 \pm 0.76^a$ | 0.4124 |
| | | GG($n = 205$) | AG($n = 269$) | AA($n = 226$) | |
| A+3262G | FEA | $278.86 \pm 1.39^a$ | $276.85 \pm 1.21^{ab}$ | $274.15 \pm 1.32^b$ | 0.0466 |
| | E59W | $69.75 \pm 1.88^b$ | $73.88 \pm 1.64^{ab}$ | $77.80 \pm 1.79^a$ | 0.0084 |
| | E300D | $19.50 \pm 1.03^b$ | $21.06 \pm 0.90^{ab}$ | $22.72 \pm 0.98^a$ | 0.0766 |
| | | GG($n = 132$) | CG($n = 231$) | CC($n = 337$) | |
| C+3301G | FEA | $272.05 \pm 1.72^b$ | $279.36 \pm 1.30^a$ | $276.42 \pm 1.08^a$ | 0.0033 |
| | E59W | $80.70 \pm 2.34^b$ | $73.11 \pm 1.77^a$ | $71.86 \pm 1.46^a$ | 0.0052 |
| | E300D | $24.77 \pm 1.28^a$ | $19.26 \pm 0.96^b$ | $21.01 \pm 0.80^b$ | 0.0028 |
| | | TT($n = 135$) | TC($n = 245$) | CC($n = 320$) | |
| T+3423C | FEA | $277.77 \pm 1.71^a$ | $278.80 \pm 1.27^a$ | $274.35 \pm 1.11^b$ | 0.0226 |
| | E59W | $67.96 \pm 2.31^b$ | $73.13 \pm 1.72^{ab}$ | $77.08 \pm 1.50^a$ | 0.0038 |
| | E300D | $20.13 \pm 1.27^{ab}$ | $19.89 \pm 0.94^b$ | $22.53 \pm 0.82^a$ | 0.0730 |
| | | TT($n = 47$) | TC($n = 154$) | CC($n = 499$) | |
| T+3428C | FEA | $270.64 \pm 2.90^b$ | $275.08 \pm 1.60^{ab}$ | $277.58 \pm 0.89^a$ | 0.0422 |
| | E59W | $85.72 \pm 3.92^a$ | $76.16 \pm 2.16^b$ | $72.14 \pm 1.20^b$ | 0.0022 |
| | E300D | $25.87 \pm 2.15^a$ | $22.16 \pm 1.19^{ab}$ | $20.38 \pm 0.66^b$ | 0.0317 |
| | | TT($n = 145$) | AT($n = 141$) | AA($n = 414$) | |
| A+3484T | FEA | $272.65 \pm 1.65^b$ | $278.96 \pm 1.67^a$ | $277.13 \pm 0.97^a$ | 0.0183 |
| | E59W | $79.86 \pm 2.23^a$ | $74.19 \pm 2.27^a$ | $71.78 \pm 1.32^b$ | 0.0080 |
| | E300D | $24.17 \pm 1.22^a$ | $19.40 \pm 1.24^b$ | $20.67 \pm 0.72^b$ | 0.0141 |
| | | GG($n = 140$) | AG($n = 218$) | AA($n = 342$) | |
| A+3489G | FEA | $278.66 \pm 1.68^a$ | $278.69 \pm 1.34^a$ | $274.35 \pm 1.07^b$ | 0.0159 |
| | E59W | $67.56 \pm 2.27^b$ | $72.51 \pm 1.82^b$ | $77.46 \pm 1.45^a$ | 0.0008 |
| | E300D | $19.99 \pm 1.24^{ab}$ | $19.56 \pm 1.00^b$ | $22.61 \pm 0.80^a$ | 0.0343 |

**Table 3** (*continued*)

| SNPs[1] | Traits[2] | Least-squares mean ± SEM[3] | | | P-value |
|---|---|---|---|---|---|
| | | GG($n = 198$) | CG($n = 180$) | CC($n = 322$) | |
| C+3545G | FEA | 272.70 ± 1.41[b] | 276.78 ± 1.47[a] | 278.81 ± 1.10[a] | 0.0029 |
| | E59W | 79.74 ± 1.91[a] | 73.11 ± 2.00[b] | 70.84 ± 1.49[b] | 0.0011 |
| | E300D | 24.26 ± 1.04[a] | 21.02 ± 1.09[b] | 19.29 ± 0.82[b] | 0.0009 |

**Notes.**

Data are summarized as means ± SEM.

[1] SNPs means single nucleotide polymorphisms, referred to covered regions, the first nucleotide of the translation start codon was designated +1, with the next upstream nucleotide being −1.

[2] FEA, first egg age; E59W, egg number at age 59 weeks; E300D, egg number at age 300 days.

[3] Values within a row with no common superscript differ significantly ($P < 0.05$) or are highly significant ($P < 0.01$).

## DISCUSSION

Muscovy duck is an excellent poultry, but its egg production is low, which has been plaguing farmers and breeders. In recent years, molecular marker breeding has gradually become the mainstream of breeding, and many breeders try to improve egg laying performances through breeding methods of molecular markers in poultry (*Wang et al., 2014a*; *Wang et al., 2014b*; *Fulton et al., 2012*; *Uemoto et al., 2009*). Using molecular marker to improve Muscovy ducks egg production is an effective method which will greatly improve the economic value of Muscovy ducks. Our study focused on egg production traits and related molecular markers, and we tried to find some molecular markers highly related to egg production in Muscovy ducks, with the hope that they can be used in Muscovy duck breeding. We believe that the relevant personnel of Muscovy ducks industry will have a strong interest in this study.

In the present study, we obtained the coding regions of *IGF2* and *DRD2* in Muscovy duck for the first time, which will be a great help in future research. *IGF2* and *DRD2* genes in humans, mice and chickens all have transcript variants (*Kaalund et al., 2014*; *Wernersson et al., 2016*; *Johannessen et al., 2016*). However, we only found one transcript in Muscovy duck. This may be caused by differences between different species.

High expression of *IGF2* in the ovary may be related to follicular development in zebra fish (*Irwin & Van Der Kraak, 2012*). In our study, we found that *IGF2* is widely expressed in different tissues with the highest expression in ovary. This suggests that *IGF2* may be associated with ovarian development. The ovarian functions of birds are regulated by luteinizing hormone (LH) and follicle stimulating hormone (FSH). *IGF2* can stimulate granule cell proliferation and related hormones synthesis and regulate follicle development with FSH in mammals (*Lucy, 2011*). Previous studies have found that *IGF2* expression in the ovary directly affects the development of dominant follicles in rats (*Wang, Asselin & Tsang, 2002*). *IGF1* can inhibit the apoptosis of granulosa cells, while *IGF2* might regulate cell proliferation during follicular development in chicken (*Johnson, Bridgham & Swenson, 2009*). In addition, *IGF2* expression in the follicles of highly productive chickens are significantly higher than that in lowly productive chickens. Therefore, a relationship exists between the expression of *IGF2* in the ovary and egg production in chickens (*Kim, Seo & Ko, 2004*). It is also becoming clear from *in vivo* and *in vitro* studies carried out in birds that *IGF2* plays an important role in ovarian follicular development (*Wood, Schlueter & Duan*,

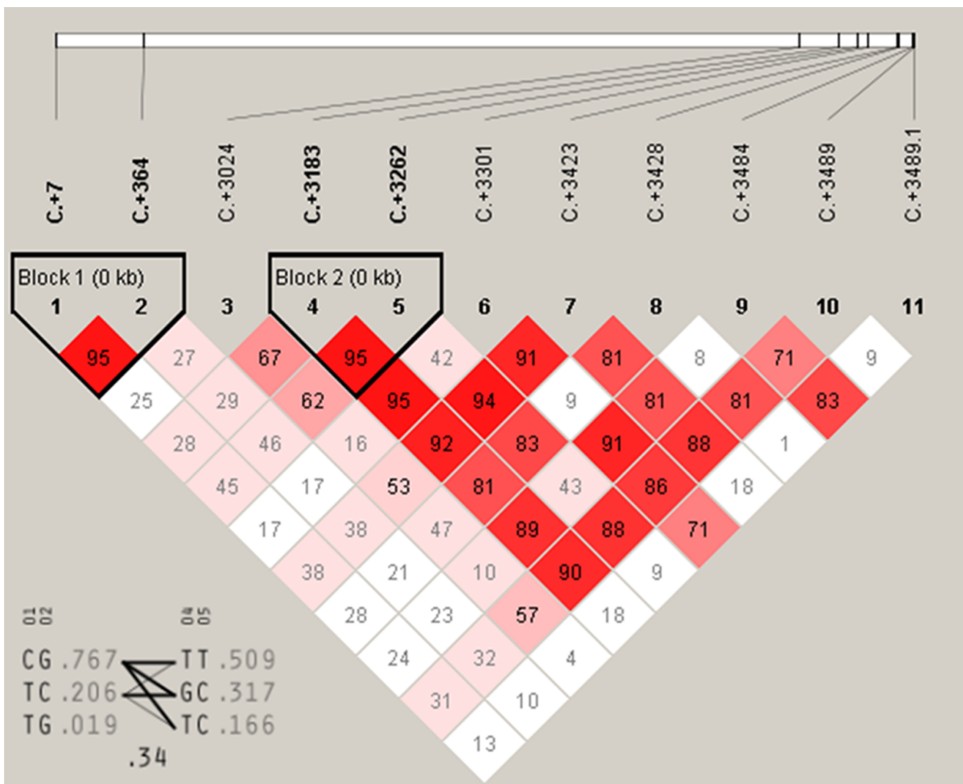

**Figure 5** **The linkage status of 11 identified SNPs in *DRD2* gene.** The color of block indicates the LD status of SNPs; deep red means high linkages between two SNPs.

*2005*). All these studies indicate that *IGF2* is related to the development of ovary. Thus, we deduced that *IGF2* might play a key role in ovarian follicular development of Muscovy ducks and regulate egg production.

In this study, we found that Muscovy *DRD2* also had its highest expression in the ovary. The *DRD2* gene belongs to the catecholamine neurotransmitter receptors that exist widely in central and peripheral nervous tissues. *DRD2* is highly expressed in the ovary and this may be related to follicular and ovarian development in humans (*Morton et al., 2006*). Other studies identified high *DRD2* expression in the regulation of reproductive functions in the grey mullet (*Nocillado et al., 2007*). DRD2 agonist can inhibit the production and secretion of vascular endothelial growth factor protein in human granulosa cells (*Ferrero et al., 2014*). Together these findings indicate that DRD2 may have a function in follicular and ovarian development. Therefore, we selected *IGF2* and *DRD2* as a candidate gene related to egg laying traits for further study.

*IGF2* is important in body growth and development. Most research on *IGF2* has concentrated on growth studies and the association of *IGF2* polymorphisms with growth related traits. Few studies have investigated the association between *IGF2* and egg laying traits. But in the current study, we found the high linkage sites A-1864G and C-1704G of *IGF2* were significantly associated with E59W. This indicated that *IGF2* was positively

related to egg laying traits. However, we have not studied how those two loci of *IGF2* regulate egg laying performance. A future study should focus on the function of the two loci for egg laying performance. Recently, *DRD2* polymorphisms have been related to poultry egg production. Our previous studies found the chicken *DRD2* gene polymorphisms were correlated with the first egg age and the egg numbers at 300 days in chicken (*Xu et al., 2011a*; *Xu et al., 2011b*). SNPs of *DRD2* were significantly associated with egg production at 38 weeks and egg weight at 300 days in chicken (*Zhu et al., 2015*). These studies suggest that the *DRD2* is indeed associated with the laying performance of birds. In our study, we also found a link between *DRD2* and the laying performance of birds. We found 10 SNPs of *DRD2* gene (C+7T, C+364G, A+3183C, A+3262G, C+3301G, T+3423C, T+3428C, A+3484T, A+3489G and C+3545G) were significantly associated with egg production traits, and two high linkage blocks were found in haplotype analysis. According to our studies, *IGF2* and *DRD2* are indeed related to the laying performance of birds, but the specific functions of these SNPs remain to be studied.

In conclusion, we identified two SNPs of *IGF2* and 11 for *DRD2*, which were highly correlated with egg laying performance in Muscovy ducks. These molecular markers highly associated with egg production traits can be used in Muscovy duck breeding. It is conducive to the development of the whole industry of Muscovy ducks. However, the functional mechanisms of these SNPs affecting egg production await further investigation.

### Funding

This study was funded by the Science and Technology Planning Project of Guangdong Province (2013B020201005), China. The funders had no role in study design, data collection and analysis, decision to publish, or preparation of the manuscript.

### Grant Disclosures

The following grant information was disclosed by the authors:
Science and Technology Planning Project of Guangdong Province: 2013B020201005.

### Competing Interests

The authors declare there are no competing interests.

### Author Contributions

- Qiao Ye performed the experiments, analyzed the data, wrote the paper, prepared figures and/or tables, reviewed drafts of the paper.
- Jiguo Xu analyzed the data.
- Xinfeng Gao, Hongjia Ouyang and Wei Luo contributed reagents/materials/analysis tools.
- Qinghua Nie conceived and designed the experiments, reviewed drafts of the paper.

## Animal Ethics

The following information was supplied relating to ethical approvals (i.e., approving body and any reference numbers):

Ethical approval for all animal experiments was granted by the Animal Care Committee of South China Agricultural University (Guangzhou, People's Republic of China) with approval number 20131019002.

## Data Availability

The raw data has been uploaded as a Supplemental File.

## Supplemental Information

Supplemental information for this article can be found online at http://dx.doi.org/10.7717/peerj.4083#supplemental-information.

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
