# Peer review of "Associations of IGF2 and DRD2 polymorphisms with laying traits in Muscovy duck"

_PeerJ, doi:10.7717/peerj.4083_

## Round 0.1 · original submission · Major Revisions

The two reviewers provided extensive constructive suggestions to improve the manuscript. Most important is inclusion of additional details on the experimental and analysis methods. Further, both reviewers emphasized the need for an expanded discussion of the overarching goals of the study and how it fits within the larger scientific context: why did you carry out the work, why should readers be interested, and what are the implications? Seek to address all of the reviewers comments, but focus particularly on these two aspects.

Reviewer 1 ·

Basic reporting

English writing still need to be improved, because there are some issues on the wording, spelling, and sentence transition and connection. For example, you should pay attention on the consistent italic style for gene names, in your manuscript, some names are italicized but some are not, please correct them.
Line 74, "Identified" change to "identified"; Line 75, changed "guide" to "reference"; Line 79, " please write full name of "dH2O".
Line 80, "3 min pre-denaturation at 94ºC" change to "pre-denaturation at 94ºC for 3 min"
Line 96, delete " mRNA" and "present", change "quantity" to "expression".
Line 106, change "sequined" to "sequenced".
Line 107, delete "using"
Line 110, change "using" to "in".
Line 116, change "egg traits' to "egg production traits".
Line 119, add "Difference with" before "P value".
Line 124-128, gene sequence of Muscovy duck is identical to "that of" Anas and Anser.
Line 137, add "also" after "other tissues".
Line 169, add "However" before "we only locate..."
Line 173, change "IGF2 high expression" to "high expression of IGF2"
Line 189, " However" change to "In addition"
Line 202, change "with" to "on".

In addition, there are some statements that lack important information for proper understanding. For example, line 32-33, IGF2 in the dominant follicles support function of which hormone? Line 70-71, cDNA was used as template to amplify what region of IGF2 and DRD2 genes? Line 110, how many females are genotyped through sequencing? Line 146, 2 SNPs of IGF2 and 11 SNPs of DRD2 were selected for further association study based on what? Line 173, IGF2 high expression in the ovary many be related to whose development? Line 181, IGF2 might regulate cell proliferation during follicular development in what animal? Line 212-213, are the two SNPs with high linkage? You never mentioned in the results part.


Literature references should also be selected carefully and properly based on the purpose of your study. In this study, your aim is to investigate association of IGF2 and DRD2 with egg production in Muscovy ducks. In this case, you should focus more on the literature about reproduction of Muscovy ducks and research of the two gene in the reproduction system of avian species, and suggest how your research is important to fill the current knowledge gaps. Why do you choose some literature on their relationship with human and fish diseases (line 35-38, line 195-199, line 204-206)? In addition, are you sure you understand the reference paper correctly? In line 197, it is DRD2 agonist used as a reagent for the treatment of the disease rather than DRD2 itself used as a agonist for the disease. I totally doubt whether your corresponding advisor has read it or not.

Experimental design

This study is a primary research within aims and scope of the PeerJ journal. However, the authors did not make enough justification for their experimental design in the introduction part. For example, why do they study Muscovy ducks instead of other birds? why do they choose IGF2 and DRD2 to study? how did they find these important genes? why do they choose E300D and E59W instead of egg production at other time points as important traits in their study?

In addition, the method is also not described sufficiently. For example, how are the primers designed for sequence cloning, qPCR and SNP detection? Although the authors listed all the primers in Table 1, they did not describe how are the primers used in each step.You should cite the primer names in the methods section and also put a note under Table 1 to explain the meaning of primer names. In addition, I suggest you to put table 1 to supplementary materials and add a figure to explain the structure of the two genes and the location of each primer in the gene sequence.

In Figure 1, you made a phylogenetic tree of muscovy duck DRD2. But you never mentioned in your method section how the tree was generated and how the sequence alignment was performed. Why do you only made phylogenetic tree for DRD2 but not for IGF2?

In Figure 2 and 3, how do the gene expressions differ among different tissues? Please label letters above each bar to denote the significant differences.

In Figure 4, you should include the whole heatmap in the figure rather than the significant block you are interested in.

In Table 2, I suggest you to add frequency of each mutation as another column.

Finally, in the animal experiment, you did not give enough information on how the animal were obtained from the company. How many ducks were offer? How were they were bred before the study? What feed were they provided?

Validity of the findings

The authors did not discussed enough to assess the impact and novelty of their study. Although the data is statistically sound and some significant SNPs were identified to be associated with egg production traits, they failed to make logical connection with current knowledge in the field, and make further biological speculations on how the SNPs may affect function of the genes and egg production traits. I think they still need work hard to improve their introduction and discussion part. Simply listing findings of previous studies in human, mammal and fish but lack of necessary rationale and linkage does not make a sound discussion at all.

Additional comments

This article described two genes and their polymorphisms, specifically high expressions in ovary, and significant SNPs related to the reproduction traits in Muscovy ducks. The study is important and interesting. However, you still need to do more work to improve English writing make it smooth to read, add more information in the methods part for sufficient description, and read more and think more to make a logical discussion to speculate possible reasons and convince the importance of your discovery. Based on the reading, I doubt whether the corresponding author has proofread and revised it or not.

Reviewer 2 ·

Basic reporting

This paper is well organized, but sustains grammatical errors.

The introduction is informative, but does not speak to the importance to this study, nor the relevance of this study to industry. Inclusion of this information would make this a more successful manuscript.

The images used in Figures 4 and 5 are of poor resolution. Using a higher quality version of this image would be advantageous.

According to the instructions for authors: multiple references listed within parenthesis should be listed chronologically; in the reference section, there should be a period between the first initial of the last author’s name and the year; the full title should be used in each of the references (see Lines 242, 250, 260, 265, among others); references with 3 authors should all be spelled out, while 4 and more use et al. and the year (these references on these lines are not cited this way in the manuscript: 255, 298, 303, 312).

The following are a sample of grammatical errors which should be addressed:
Line 32 – Especially is not the correct word to use in this sentence. It should be removed.
Line 58 – First egg age acronym not consistent as compared with the rest of the manuscript.
Line 117 – statement should state “effect of each genotype”; genotype should be singular
Line 120 – a space should appear before and after the less than or equal sign.
Line 149 – should read “C-1704G SNP of the IGF2 gene”
Lines 154 & 159 &160– “Highly significantly” should be replaced with “indicate highly significant associations”
Line 157 – Should read “highly associated with E59W”
Line 172 – “Was” should be replaced with “is”
Line 172 – “but predominantly” should be replaced with “with the highest expression found”
Line 181 – add a comma before “while”
Line 183 – should read “highly productive”
Line 183 – change “were” to “are”
Line 184 – change “ones” to “chickens”
Line 266 – remove extra space
Line 352 – Should read “or are highly significant”

Experimental design

Answering the following questions would provide clarity to the manuscript:
Why were egg collection data taken specifically at 300 days and 59 weeks?
Where the ducks all raised in the same run, or were portions raised at different time points/seasons/locations?
What age were the ducks euthanized?
Where the same ducks used to gather production data and tissue data for expression analysis?
Why were some SNPs chosen over others for association studies?

Validity of the findings

Be specific when defining the tissues expression of IGF2 occurs in in the Tissue Expression of IGF2 and DRD2 Genes section.

The explanation of Table 3 in the Results section should be re-assessed to make it more clear. The superscripts differentiating AA, AG, and GG of A-1864G for the E59W trait should be listed as a, ab, and b, respectively. The p-values reported in the text, and those in the tables are different. Clarifying the reasoning behind this would make this paragraph much more understandable.

The presence or absence of associations in T+3024C and A+3183C for DRD2 should be addressed in the Results section.

After reading this manuscript in its entirety, it remains unclear the purpose of this study, and implications of its findings relevant to the duck industry. This information would make this paper much stronger. Additionally, the conclusion does not explore reasonings behind association of certain SNPs to specific production parameters measured, more than a just a repeat of basic findings from the Results section. This manuscript would greatly benefit from a re-written conclusion that goes further into interpreting results from this study.

Concerns should be raised about the validity of the findings in the expression analysis, since age of tissue collection is not explained. The information does not appear to help add to the findings of the manuscript. At which point in the duck’s production timeline tissues are collected could play a major role in the expression levels. This information would be more useful if gathered at the same time points the production traits were collected (AFE, E300D, E59W). Having both SNP association data and expression data would present a more informative discussion as to the results in this manuscript.

Additional comments

Information regarding functions of IGF2 are reported in mammals, but would be more compelling if linked to poultry, or a statement should reflect that the author could not find any such evidence in poultry to clarify.

Table 4 superscripts are not consistent in capitalization.

Listing the table number within the first sentence of the explanation of the results would improve the flow of the findings.

In two cases (Zhang on lines 38 and 42; Ferrero on lines 198 and 200) one of each of these author’s references were not correctly cited in the text, therefore, there is no way to distinguish which reference is being referred to.

---

## Round 0.2 · Minor Revisions

Please pay particular attention to the comments of Reviewer 2. The introduction needs to be expanded further, to make the rationale for the study more clear. The introduction is currently a single, one-page paragraph; I think it needs to be made at least twice as long. Why study this topic, what was known previously, what new information has this study provided?

Also add more explanations in the Methods sections regarding the rationale for the various experimental decisions, as requested by Reviewer 2.

The superscripts in Table 3 are not fully explained. I take it that the lower and upper case superscripts concern P<0.05 and P<0.01. This seems unnecessarily complicated. Perhaps just include one or the other, and maybe add a paragraph in the text explaining what these mean.

Personally, I'd prefer to see Tables 2 and 3 replaced with figures. I think the strength of the associations would be more clear.

Reviewer 1 ·

Basic reporting

English writing can still be improved.
In introduction section:
Line 2: change "prized" to "priced".
Line 5: change "pay" to "paid"
Line 8: E300D and E59W are both egg production traits at certain time age, how can they be peak time and end time for laying. Please revise this sentence.
Line 9: add "Therefore," before "we"

In sample collection part of Material and Methods section:
Line 1: change "800" to "Eight hundred"
Line 2: add "were" between "which" and "in"

change "distillation-distillation" to "double distilled"

In Polymorphisms of IGF2 and DRD2 Genes part:
change "S" to "s" in "sequencing"

In Association of IGF2 and DRD2 with egg production traits part:
Line 2: Change "Linkage disequilibrium" to "linkage disequilibrium analysis"
Line 8: change "were" to "had", delete "indicate"
Line 13: change "were" to "was"

In discussion section:
Line 1: delete space in "poultry", change "it" to "which"
Line 2: change "i" to "I" in "improving"
Line 5: change "I" to "We"
Line 14: delete "found", change "suggest" to "suggests"
Line 22: add "that" after "than", change "low productivity" to "lowly productive"
Line 34: It is "DRD2 agonist" rather than "DRD2" that "the production and secretion of vascular endothelial growth factor protein in granulosa cells"

Experimental design

The experiment design is good and enough to answer the research question. Methods is also described with sufficient details and information.

Validity of the findings

Data in this study is robust and statistically sound. Reasonable conclusion and speculations are also made based on the results.

Additional comments

Generally, the manuscript have been greatly improved after your revision. But English writing still needs to be improved. I believe it is a good study that would be published soon.

Reviewer 2 ·

Basic reporting

Content added by the authors in the Introduction, Methods, and Discussion sections contain substantial grammatical and syntax errors in the revised manuscript. The information included did not aid in clarifying the relevance of this study. This manuscript contains potentially important information for the duck industry, but the authors have not adequately conveyed this message in these sections. Additionally, without appropriate citations to back up their claims which are used to substantiate the significance of this study, overall the argument is less compelling.

Table 3 remains hard to understand. The current title is not sufficient in allowing the reader to understand everything going on, and does not make clear how to differentiate the meaning between lower case and capitol letter associations in the LEM.

Despite mention of the correct formatting for references in the initial review comments, two-authored references are not cited correctly in the text of the manuscript. Finally, sentences ending in the citation of multiple references are not listed chronologically.

Experimental design

A brief statement as to the rationale of eggs at day 300 and week 59 is warranted, even if widely known in the duck production community. PeerJ readers are likely to not know of the specifics of duck egg laying cycles.

The sampling methods indicate in the manuscript that all ducks were run together, while in the rebuttal, the author states that ducks sampled for production and tissue analysis were from a “different batch”.

The rationale behind selecting 2 out of the 5 SNP identified in IGF2, and 11 out of 28 SNP identified in DRD2 needs to be explained in the text.

Validity of the findings

The impact of this study was not sufficiently addressed in the revision. The discussion includes statement in reference to non-avian species, but fails to acknowledge whether any information is known within birds, or if positive associations of the data are being backed up by the authors for different taxa.

The authors did a good job of noting that only positive associations were identified, but that further investigation is warranted in regards to causation due to the SNPs

---

## Round 0.3 · Minor Revisions

Your revised manuscript does a great job of accommodating the reviewers' suggestions.

I have just a few small changes to suggest.

My main comment: I think it would be best to use a log scale for Figures 2 and 3. I'm sorry I didn't mention this previously, but gene expression values (which vary over many orders of magnitude) are generally best viewed on a log scale.

Minor comments

1. Line 31, page 2: I think "highly priced meat" should actually be "highly prized meat" as you had it in the previous version. Or maybe just change it to "high quality meat".

2. Include a citation for Haploview: (see https://www.broadinstitute.org/haploview/haploview)

Barrett JC, Fry B, Maller J, Daly MJ. Haploview: analysis and visualization of LD and haplotype maps. Bioinformatics. 2005 Jan 15 [PubMed ID: 15297300]

3. I attach a marked-up PDF of your manuscript with many minor wording suggestions.

---

## Round 0.4 · accepted · Accept

There are a number of typos related to insertions you made: lack of space between words:

line 56 "report that"
line 146 "using a"
line 179 "that the"
line 190 "associated with"
line 191 "the three"
line 206 "that they"
line 208 "have a"
line 246 "future study"